# Neutrophil-to-Lymphocyte Ratio and Platelet-to-Lymphocyte Ratio as Predictive Factors for Mortality and Length of Hospital Stay after Cardiac Surgery

**DOI:** 10.3390/jpm13030473

**Published:** 2023-03-05

**Authors:** Georgios Tzikos, Ioannis Alexiou, Sokratis Tsagkaropoulos, Alexandra-Eleftheria Menni, Georgios Chatziantoniou, Soultana Doutsini, Theodosios Papavramidis, Vasilios Grosomanidis, George Stavrou, Katerina Kotzampassi

**Affiliations:** 1Department of Surgery, Aristotle University of Thessaloniki, 546 36 Thessaloniki, Greece; 2Department of Cardiothoracic Surgery, Aristotle University of Thessaloniki, 546 36 Thessaloniki, Greece

**Keywords:** cardiac surgery, mortality, neutrophil-to-lymphocyte ratio (NLR), platelet-to-lymphocyte ratio (PLR), prognostic factor

## Abstract

Background: Neutrophil-to-lymphocyte ratio (NLR) and platelet-to-lymphocyte ratio (PLR) are widely accepted indices positively correlated with disease severity, progression, and mortality. In this study, we tested whether NLR and PLR could predict mortality and length of hospital stay (LOS) after cardiac surgery. Methods: NLR and PLR were calculated on days 0, 3, 5, and 7 postoperatively. A ROC curve was generated to assess their prognostic value; multivariate logistic analysis identified independent risk factors for 90-day mortality. Results: Analysis was performed on 179 patients’ data, 11 of whom (6.15%) died within 90 days. The discriminatory performance for predicting 90-day mortality was better for NLR7 (AUC = 0.925, 95% CI:0.865–0.984) with the optimal cut-off point being 7.10. NLR5 and PLR3 also exhibited a significant strong discriminative performance. Similarly, a significant discriminative performance was prominent for PLR3, NLR5, and NLR7 with respect to LOS. Moreover, NLR7 (OR: 2.143, 95% CI: 1.076–4.267, *p* = 0.030) and ICU LOS (OR:1.361, 95% CI: 1.045–1.774, *p* = 0.022) were significant independent risk factors for 90-day mortality. Conclusions: NLR and PLR are efficient predictive factors for 90-day mortality and LOS in cardiac surgery patients. Owing to the simplicity of determining NLR and PLR, their postoperative monitoring may offer a reliable predictor of patients’ outcomes in terms of LOS and mortality.

## 1. Introduction

Complete blood count (CBC) is one of the most common examinations performed in the everyday clinical setting to provide information about red blood cells, white blood cells and their subtypes, and platelets. Furthermore, the indices derived from them such as the neutrophil-to-lymphocyte ratio (NLR), platelet-to-lymphocyte ratio (PLR), monocyte-to-lymphocyte ratio (MLR), and neutrophil-to-platelet ratio (NLP) have been found to be well correlated with disease severity, progression, and mortality [1], reflecting the immune system’s sufficiency and its ability to respond to inflammation. The cellular-mediated inflammatory response includes neutrophils, lymphocytes, monocytes, and platelets; the indices derived from them seem to be reliable biomarkers of the immunological status of the patient and, thus, predictive factors for the disease progression [2], as have been proven both in ICU trauma patients and in a variety of malignancies [3,4,5,6,7,8,9,10,11,12].

Patients undergoing anesthesia for cardiac surgery experience a systemic inflammatory response more aggravated than in other surgeries due to the additional stress of extracorporeal circulation [13,14]. A variety of hematological indices has been referred to in the literature as having different predictive powers in different clinical situations; perioperative NLR and PLR, used separately or as a pair, have been tested for the prognostication of postoperative morbidity and mortality, as well as for severe postoperative complications in cardiac surgery patients with sometimes good and sometimes ambiguous results. These indices, to some extent, reflect the extent of the inflammatory response occurring after the stress of an operation [15,16].

This study aims to assess the potential predictive value of postoperative NLR and PLR in patients undergoing cardiac surgery regarding 90-day postoperative mortality and length of hospital stay.

## 2. Materials and Methods

### 2.1. Study Design and Population

This is a post hoc analysis of data collected prospectively for the primary study assessing the prognostic validity of nutritional status, body composition, phase angle, and muscle strength regarding morbidity and mortality in a cardiac surgery population from September 2018 to August 2019 [17]. All the patients scheduled for elective coronary artery bypass grafting or isolated valve replacement or repair, by means of minimally invasive extracorporeal circulation, were eligible for the primary study. 

The exclusion criteria were: 1. age < 18 years old, 2. clinical instability necessitating emergency surgery, 3. congenital heart abnormality, 4. aortic dissection, 5. recent (≤3 months) open heart surgery, and 6. the existence of an implantable electronic cardiovascular device. The patients were admitted one day before the surgical intervention and completed the mandatory preoperative assay. Written informed consent was obtained from all subjects; the study protocol was approved by the authors’ institutional review board and prospectively registered with ClinicalTrials.gov (NCT03644030). This work has been reported in line with the STROCSS criteria [18].

### 2.2. Data Collection

Mortality was defined as death occurring from any cause during the first 90 days after cardiac surgery. After the sample size had been recruited, general data, such as sex, age, demographics information, weight, height, behavioral history (smoking, alcohol, and drug use), preoperative diagnosis, any accompanied comorbidities, EuroSCORE II, ejection fraction (EF), MUST score for the assessment of nutritional status, postoperative records for length of ICU and hospital stay, and follow-up laboratory exams were obtained from medical records of the selected patients. 

After having retrieved the data from the patients’ general blood tests, the values of lymphocytes, neutrophils, and platelets preoperatively and on the 3rd, 5th, and 7th postoperative days were used for NLR and PLR calculations. The NLR ratio was defined as the absolute neutrophil count divided by the absolute lymphocyte count on a given day, and the PLR as the absolute platelet count divided by the absolute lymphocyte count. These parameters were named NLR0, NLR3, NLR5, and NLR7 as well as PLR0, PLR3, PLR5, and PLR7, for day 0 (preoperatively) and postoperative days 3, 5, and 7, respectively. 

### 2.3. Statistical Analysis

Statistical analysis was conducted with the Statistical Package for Social Science (SPSS), Inc. (v 25.0; Chicago, IL, USA). The normality of the data’s distribution was assessed using the Kolmogorov–Smirnov or Shapiro–Wilk test when the evaluated group included more or less than 50 patients, respectively. For continuous variables, the results were presented as mean ± standard deviation (SD) if normality was assumed, and as the median and the interquartile range (IQR) for variables with skewed distribution. Moreover, for the comparison of two independent samples’ means, an independent sample t-test was performed, while the Mann–Whitney test was used to find the differences between the medians of two independent samples. Qualitative data were presented as percentages and the chi-square test was used for nominal variables. Multivariate logistic regression was performed for identifying the potential predictive factors responsible for 90-day mortality. Thus, after the univariate logistic analysis of every possible factor had been performed, a multivariate logistic analysis was conducted, and the final model was built. During univariate regression, a factor was included in the multivariate logistic regression model when it met a statistical significance of a *p* value less than 0.20. The final model was built using a stepwise backward elimination method with a significance level of 0.05. In addition, a receiver operating characteristic (ROC) curve was generated to calculate the optimal cut-off point of the derived indices for 90-day mortality and length of hospital stay, which was chosen based on the accompanying Youden’s index [19,20]; sensitivity and specificity were also measured. After the optimal cut-off points for NLR and PLR had been calculated, the sample size was divided into two groups based on them, and the mortality incidence was reassessed. Any linear correlation was assessed with the help of the Pearson or Spearman correlation coefficients, when normality was or was not assumed, respectively. Finally, a post hoc power analysis was performed to estimate the power of this study [21].

## 3. Results

One hundred eighty-nine consecutive patients were eligible for this post hoc analysis. Ten were lost to follow-up and there were no available data from their records. Thus, 179 were included in the final analysis. The post hoc power analysis for the NLR7 [mortality (20%) when NRL7 > 6.60 versus mortality of the whole sample (6.15%)] revealed a power of 81.2% with a level of significance of 0.05 (two-sided). The demographic and clinical characteristics of the participants are presented in Table 1.

The diagrams of hematological parameters and the derived indices as they varied up to postoperative day 7 are shown in Figure 1. 

Additionally, the data from the blood analysis and the derived hematological indices are presented in Table 2. Eleven patients died within the first 90 postoperative days (rate:6.15%). The time to death measured in days has a mean of 32.9 days (SD = 19.5 days) and a median of 29 days (min = 15 days, max = 85 days). Six patients died within the first month postoperatively and the other five during the second and the third month. There was no difference between survivals and non-survivals regarding the duration of extracorporeal circulation (78.2 ± 32.5 min vs. 77.2 ± 28.7 min, *p* = 0.921).

There is no sign of difference, or, in other words, no predictive value of NLR0 and PLR0 for patients who finally died (*p* = 0.185 and *p* = 0.915, respectively). The mean time of patients being intubated was 1.52 ± 2.20 days (median: 1.0, min: 1.0, max: 26.0) for survivals and 10.72 ± 9.03 days (median: 12.0 days, min: 1.0, max: 25.0) for non-survivals (*p* < 0.001). Eight out of 11 patients who died remained intubated and under ventilatory support for more than 48 hours. Thus, we continued testing the predictive value of NLR and PLR on the 3rd, 5th, and 7th postoperative days and found that only NLR5, NLR7, and PLR3 showed significantly good performance.

### 3.1. NLR and 90-Day Mortality

Our findings highlighted the NLR7 values as the most predictive factor for 90-day mortality. The probability of death was greater as the NLR value increased. As shown in Figure 2, the discriminative performance for predicting 90-day mortality was better for NLR7 (AUC = 0.925, CI 95%: 0.865–0.984) than for NLR5 (AUC = 0.810, CI 95%: 0.678–0.942). The optimal cut-off point was 6.60 for NLR7, with a sensitivity of 100.0% and specificity of 78.2%, while for NLR5, it was 7.10, with a sensitivity of 100.0% and specificity of 50.0%.

When patients were divided according to the optimal cut-off point of 6.60 for NLR7, the mortality was found to be 20.00% (11 out of 55 patients) in patients with an NLR7 value > 6.60, which was significantly higher compared to the survival of the rest of the patients (0%, *p* < 0.001) and that the whole sample overall (20.00% vs. 6.15%, *p* = 0.002) as well. Similarly, after patients had been divided according to the optimal cut-off point of 7.10 for NLR5, the mortality was 10.52% (10 out of 95 patients) in patients with an NLR5 value > 7.10, which was also significantly higher compared to the mortality of the patients with NLR5 value ≤ 7.10 (1.19%, *p* = 0.011), but not compared to the overall survival of the sample (10.52% vs. 6.15%, *p* = 0.194).

### 3.2. PLR and 90 Day Mortality

Regarding PLR, as opposed to NLR, the probability of death increases as the PLR value decreases. Its discriminatory performance for predicting 90-day mortality after cardiac surgery and the respective ROC curve is presented in Figure 3. More specifically, the PLR3 exhibited a significant strong discriminatory performance (AUC = 0.714, CI 95%: 0.581–0.847), while PLR5 and PLR7 values were not statistically significant (PLR5: AUC:0.585, 95% CI: 0.371–0.798, PLR7: 0.650, 95% CI: 0.414–0.887). Regarding the optimal cut-off point of PLR3, this was 126.34, with a sensitivity of 100.0% and specificity of 43.7%. Based on these, it should be mentioned that although PLR5 and PLR7 failed to be discriminative predictors, 9 out of the 11 patients who died had not only PLR3 < 126.34 but also NLR5 > 7.10 and NLR7 > 6.60. Additionally, after splitting the patients according to the PLR3 cut-off point of 126.34, it was found that in those 99 patients having low PLR3 values, the mortality rate was significantly higher (10.1%) compared to that (1.25%) of the remaining 80 patients (*p* = 0.017), but not compared to the overall survival of the sample (*p* = 0.194).

### 3.3. NLR, PLR, and 90-Day Mortality

For further analysis, we divided the patients into four categories based on an arbitrary point system according to the number of predictors that were met. We considered “abnormal” regarding NLR any NLR5 value > 7.10 and any NLR7 > 6.60, and for PLR, any PLR3 value < 126.34. Thus, four categories were formed: 0: when neither PLR3, NLR5, nor NLR7 were abnormal, 1: when one of PLR3, NLR5, or NLR7 was abnormal, 2: when two of them were abnormal, and 3: when all of them were abnormal. The results are presented in Table 3; it is clear that the mortality rate was 0% for the 107 patients having a score of 0 or 1, 4.35% for the 46 patients having a score of 2, and 34.6% for the 26 patients having a score of 3 (*p* < 0.001).

Then, a multivariate binary logistic regression analysis was performed in order to identify any potential independent risk factors for mortality. In the univariate analysis, we included, besides the factors shown in Table 4, PLR3, PLR5, and NLR7 (as scale variables), which had shown good discriminatory performance in the ROC curve analysis. In the multivariate analysis, we included gender, age, and MUST score (regardless of the fact that they have not achieved the level of significance we have set (*p* = 0.20) for making a factor eligible for inclusion in the multivariate analysis) because of their clinical significance for heart diseases. We also included EuroSCORE II, ICU length of stay, PLR3, NLR5, and NLR7, which achieved a significance level of less than 0.20. The multivariate analysis finally revealed that only NLR7 (OR: 2.143, 95% CI: 1.076–4.267, *p* = 0.030) and ICU length of stay (OR:1.361, 95% CI: 1.045–1.774, *p* = 0.022) were significant independent risk factors for 90-day mortality after cardiac surgery, adjusted for gender, age, and MUST score, while PLR3 and NLR5 lost their significance. The detailed results are presented in Table 4.

### 3.4. NLR, PLR, and Length of Hospital Stay

Regarding the predictability of NLR and PLR for the prolongation of in-hospital length of stay (>7 days), PLR3, NLR5, and NLR7 were found to have the best-fitted discriminative performance. The respective ROC curves are presented in Figure 4. 

The AUC for PLR3 was 0.616 (CI 95%: 0.516–0.717) and the optimal cut-off point 131.44, with a sensitivity of 66.9% and specificity of 55.3%; for NLR5, the AUC was 0.637 (CI 95%: 0.535–0.738) and the optimal cut-off point 7.93 with a sensitivity of 50.0% and specificity of 78.4%; and for NLR7, the AUC was 0.638 (CI 95%: 0.544–0.732) and the optimal cut-off point 4.03 with a sensitivity of 60.5% and specificity of 74.3%f. PLR3 and length of hospital stay exhibited a negative linear correlation, with a Spearman correlation coefficient of −0.255 (*p* = 0.001). Finally, when the PLR and NLR values from the 3rd, 5th, and 7th postoperative days were assessed for the potential of a linear correlation, significant strong evidence was only found between PLR3 and NLR3 (Table 5). 

## 4. Discussion

In a recent systematic review and meta-analysis of 12 studies with 13,262 cardiac surgery patients, Perry et al. [22] conclude that the perioperative NLR value is an independent predictor of short-term and long-term postoperative mortality, besides a considerable between-study statistical heterogeneity (I^2^ = 94.39%) explained by the study-level variables. In light of their findings, we decided to perform the present post hoc analysis of data collected from our group for a primary cohort survey [17], where we have shown that cardiac surgery patients are at risk of nutritional status deterioration—as assessed by means of BIA, mainly phase angle and fat-free mass—positively related to morbidity and mortality; our data regarding NLR and PLR were, thus, evaluated as predictive indices for 90-day mortality, trying to avoid the bias reported in the aforementioned meta-analysis. 

Among all the parameters analyzed, the NLR5 and NLR7, as well as PLR3, were found to exhibit good discriminatory performance for predicting 90-day mortality. The multivariate analysis performed thereafter showed that NLR7 and the ICU length of stay were independent risk factors for death, adjusted for age, gender, and MUST score. Prolonged ICU length of stay has also been confirmed as an independent risk factor in other studies, which reported that postoperative morbidity and mortality are increased in patients with prolonged ICU stay after cardiac surgery. Moreover, prolonged hospitalization is mainly associated with respiratory events and prolonged ventilator intubation time [23]. Thus, in our study, most of the patients who finally died had remained intubated for more than 48 hours in the ICU. At this point, it should also be noted that in the multivariate analysis model, the preoperative MUST score, which is an indicator of patients’ nutritional status, was not associated with increased mortality, because there was no difference detected in the median MUST score between the patients who died and those who survived.

Furthermore, when dividing the patients according to “points” received—one point per each positive predictor NLR5, NLR7, and PLR3—most of the patients having the highest score of 3 points eventually died (9 out of 11). Moreover, PLR3, NLR5, and NLR7 also have good performance for predicting the prolongation of hospital stay of more than 7 days. 

It is well known that the contribution of NLR and PLR in evaluating the immune status of patients, and consequently their inflammatory response, has been studied thoroughly in the last two decades [24]. Moreover, during an overwhelming inflammatory response, lymphocytopenia and lymphocyte hypoactivity occur, due to B-cells and T-cells apoptosis, both contributing to greater mortality [25,26]. In addition to lymphocytopenia, neutrophilia and inappropriate systemic neutrophil activation and migration within the microvasculature contribute to tissue damage and multiple organ failure [27]. Furthermore, in cardiac surgery patients, besides the operational stress itself, the additional use of the stressful extracorporeal circuit triggers an unavoidable major immune response, accelerated through the contact of the blood products and the surfaces of the CPB tubes. Additionally, during anesthesia and surgery, the activation of the neuroendocrine system results in the release of cytokines and hormones, which induce systemic leukocyte alterations affecting NLR [28]. In our study, the mean cardiopulmonary bypass time was 78.0 ± 32.3 min, and although the aortic cross-clamp time was not recorded, it was estimated to be around 20% less than the time needed for extracorporeal circulation. 

NLR and PLR have been evaluated, either separately or together, as predictive factors for mortality related to cardiovascular outcomes [29,30]. The prognostic performance of these indices has been also evaluated by others on cardiac surgery patients [31,32,33,34]. In a recent retrospective study of 1694 patients divided into two groups according to their preoperative NLR optimal cut-off point of 3.23, the authors conclude that patients having an NLR value greater than 3.23 experienced greater mortality (OR:3.36, 95% CI: 1.63–6.91); other parameters such as the ICU stay were also affected [35]. The predictive value of preoperative NLR and PLR values for major adverse cardiovascular and cerebrovascular events has also been confirmed by Larmann et al. [36], who reported a cut-off point of >204.4 for PLR and >3.1 for NLR. However, it should be highlighted that both studies referred to the preoperative NLR and PLR values, this probably being the reason they report lower cut-off values compared to our results. In our study, we evaluated the postoperative values of NLR and PLR, which are definitely affected by the inflammatory response elicited by operational stress. Very similar are the results of Zhu et al. [37], who found the critical postoperative NLR value to be 7.23 when they correlated NLR with mortality after cardiac surgery. In the same manner, the optimal cut-off point for predicting mortality with NLR on postoperative day 1 was 7.28 in a recent retrospective study including 2707 cardiac surgery patients. 

Our post hoc analysis also reveals that the lower the PLR value, the higher the possibility of death. This could be easily explained by the fact that thrombocytopenia is a sign of both infection and inflammatory response; platelets interact with white blood cells or vascular endothelial cells directly, based on a contact-dependent mechanism, and indirectly through the secretion of inflammatory cytokines [38]. Thus, the involvement of platelets in the inflammatory process is noted both locally and systemically [39]. This recruitment of platelets in combination with their adhesion to white blood cells to enhance their effect reduces the absolute number of circulating platelets, which is then reflected in the decrease in PLR [40]. This is why we demonstrate a reversed probability—the lower the PLR, the higher the probability of death—while others, by using preoperative values, support a positive correlation [36]. 

It is of interest to mention that when the patients were divided according to the points received—one point per each positive predictor NLR5, NLR7, and PLR3—we found that 9 out of 11 patients who died presented with all the parameters (PLR3, NL5, and NLR7) positive, whereas when none of the indices were abnormal, all the patients survived. This finding is in accordance with a previous study, where the predictive power increased in parallel with the number of abnormal parameters measured (NLR, red cell distribution width, and mean platelet volume) [15]. Based on all the aforementioned findings, we could support the suggestion that these parameters could be highly informative for the postoperative monitoring of cardiac surgery patients, and they are easy to perform and of a very low cost; they consist of a routine, daily practice, and calculating them during the first 7 postoperative days may let the patients who are at higher risk for adverse outcomes, including mortality, to be detected, even from the 3rd postoperative day.

It is commonly accepted that doctors have a perpetual desire and will to be able to predict the prognosis of their patients. However, this commitment should not let them forget that their role is to provide the best possible care to critically ill patients, regardless of whether the odds are in their favor or whether the patient’s survival would be a minor miracle. Healthcare professionals should follow the “deontological theory” and try to gain the greatest good for the patient and act for the patients’ benefit [41]. Given the above, the strategy of predicting as soon as possible which patients have a higher chance of experiencing worse morbidity or higher mortality aims at triggering the reflexes of physicians in time and alerting them to possible complications that may arise in the future. In this way, it will be possible to ensure maximum patient-centered medical care delivery with the lowest possible consumption of resources, as these are consumed on the basis of need rather than horizontally. 

The contribution of the NLR and PLR predictive value to personalized medicine is pivotal because these easily CBC-derived indices are useful information that could affect treatment decisions about patients. These include identifying patients at higher risk of experiencing a more stressful postoperative course after major cardiac surgery; those patients should be monitored more closely and carefully. Additionally, NLR and PLR in combination with other equally simple and available biomarkers, such as C-reactive protein, erythrocyte sedimentation rate, or even procalcitonin and pancreatic stone protein, could take a “one size fits all” treatment approach and turn it into an individualized and patient-centered approach, which would be more effective and possibly less costly.

There are some limitations in this study. First, it is a post hoc analysis of retrospectively collected data for the primary work. Second, based on the variation in hematological parameters, our sample size could be considered small. However, the conducted post hoc power analysis has revealed a power of 81.2%; thus, the results are reliable, and safe conclusions can be drawn. Finally, regarding the mortality observed in our study, 30-day mortality (6 out of 179 patients) was 3.35% and 90-day mortality (11 out of 179 patients) was 6.15%. Although these percentages could be considered relatively high, they are equivalent to the percentages presented in the original EuroScore II study (4.048% and 6.023%, respectively) [42], while in this study, we aimed to evaluate only the predictive character of the aforementioned hematological parameters and, thus, analyzing the causes of these results would be beyond the purposes of our work. 

## 5. Conclusions

The present post hoc analysis strengthens the suggestion that NLR and PLR values are efficient predictive factors for 90-day mortality and hospital length of stay in adult cardiac surgery patients. More specifically, NLR5, NLR7, and PLR3 seem to exhibit the best discriminatory values for predicting 90-day mortality, whereas NLR5 > 6.60, NLR7 > 7.70, and PLR < 126.34 were found to be the optimal predictive cut-off points. Additionally, an elevated postoperative NLR value (NLR7 > 6.60) and the ICU length of stay are independent risk factors for increased 90-day mortality. Additionally, more than two of these indices being positive significantly increases the possibility of death occurring. The aforementioned findings can be simply summarized with the perspective that, owing to the simplicity of determining the NLR and PLR values, monitoring them can predict the early outcome of adult patients undergoing cardiac surgery.

## Figures and Tables

**Figure 1 jpm-13-00473-f001:**
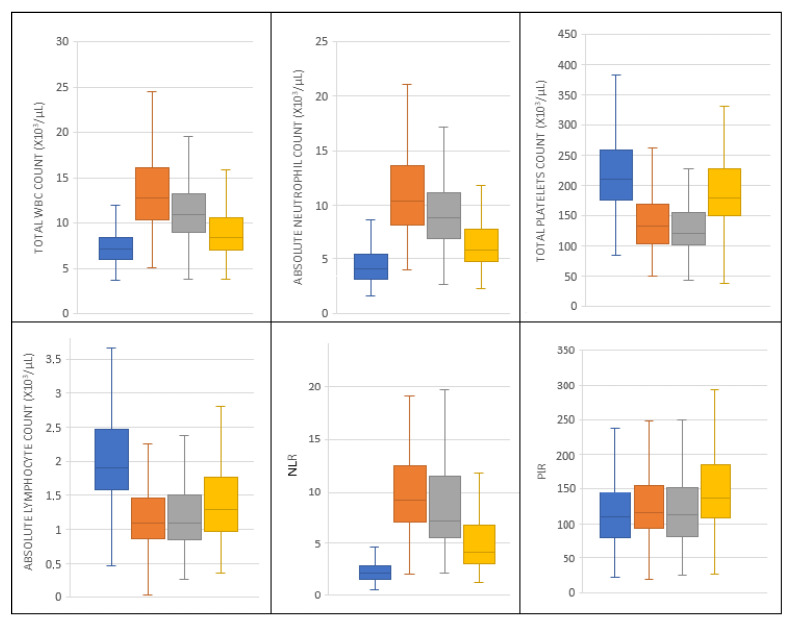
The fluctuations of hematological parameters and their derived indices preoperatively (blue) and on postoperative day 3 (orange), 5 (grey), and 7 (yellow).

**Figure 2 jpm-13-00473-f002:**
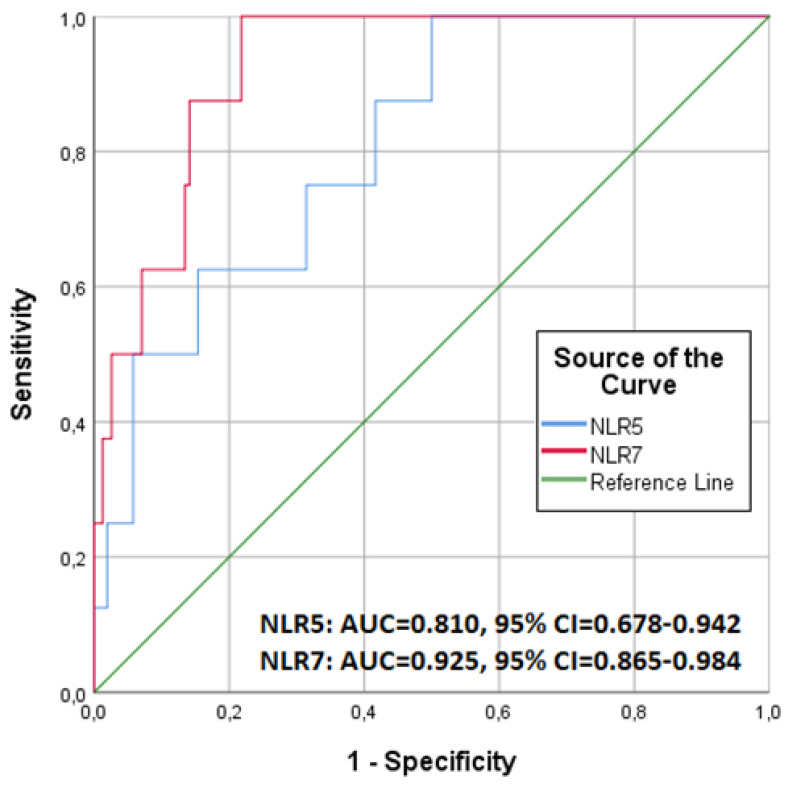
Receiver operating characteristic (ROC) curves for predicting 90-day mortality for neutrophil-to-lymphocyte ratio (NLR) calculated on the 5th (NLR5) and the 7th (NLR7) postoperative day.

**Figure 3 jpm-13-00473-f003:**
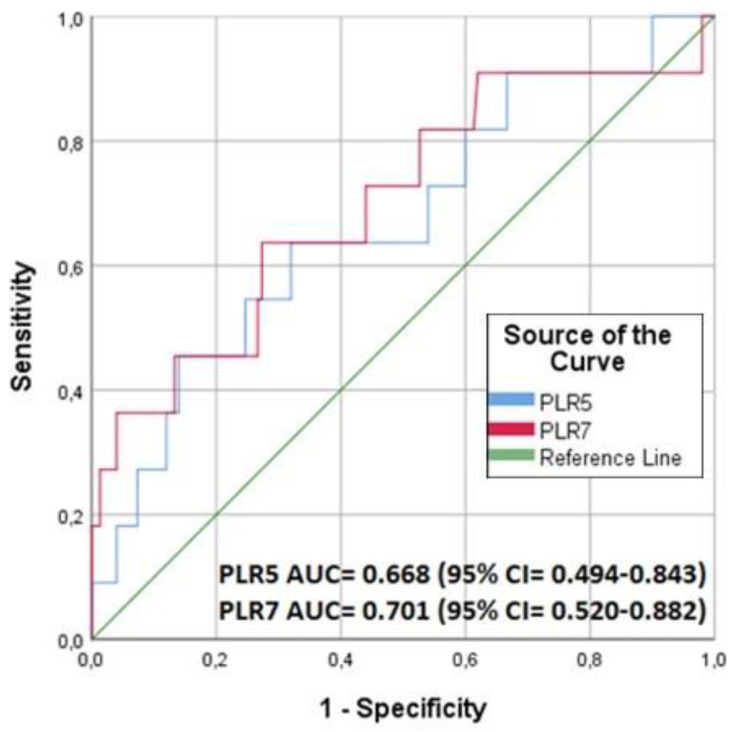
Receiver operating characteristic (ROC) curves for predicting 90-day mortality for platelet-to-lymphocyte ratio (PLR) calculated on the 5th (PLR5) and the 7th (PLR7) postoperative day.

**Figure 4 jpm-13-00473-f004:**
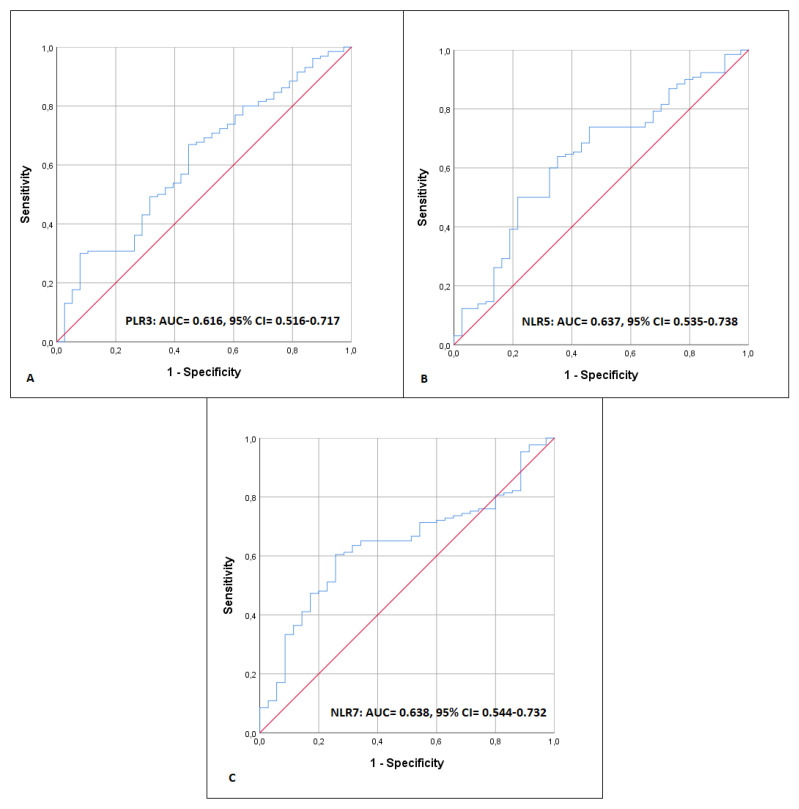
Receiver operating characteristic (ROC) curve for predicting length of hospital stay more than 7 days for platelet-to-lymphocyte ratio (PLR) calculated on the 3rd (PLR3) postoperative day (**A**), and neutrophil-to-lymphocyte ratio (NLR) calculated on the 5th (**B**) and 7th (**C**) postoperative day (NLR5, NLR7).

**Table 1 jpm-13-00473-t001:** Demographic and clinical characteristics of the participants.

N = 179	Coronary Artery Disease		Heart Valvular Disease		Total		*p*-Value
Patients (n)	90		89		179		n/a
Gender (male)-n (%)	78 (86.7%)		61 (68.5%)		139 (77.7%)		0.004
Age (years)	68.5	(13.0)	67.0	(15.0)	68.0	(13.0)	0.471
Height (meters-m)	1.69	(0.10)	1.70	(0.11)	1.70	(0.10)	0.524
Weight (kilograms-kg)	77.2	(19.1)	78.3	(19.2)	77.6	(18.5)	0.824
MUST Score	1.0	(1.0)	1.0	(1.0)	1.0	(1.0)	0.251
BMI (m/kg^2^)	26.4	(5.1)	27.8	(6.4)	27.2	(5.8)	0.308
EuroSCORE II (%)	2.1	(2.1)	3.0	(1.8)	2.4	(2.0)	0.022
Ejection Fraction (%)	50	(10)	55	(5)	55	(5)	0.007
ICU Stay (days)	1.0	(1.0)	2.0	(2.0)	2.0	(2.0)	0.738
Ward Stay (days)	7.0	(2.0)	7.0	(2.0)	7.0	(2.0)	0.115
Total Hospital Stay (days)	8.0	(3.0)	9.0	(3.0)	9.0	(3.0)	0.670
Diabetes Mellitus-n (%)	44 (48.9%)		38 (42.7%)		82 (45.8%)		0.894
COPD-n (%)	27 (30.0%)		25 (28.1%)		52 (29.1%)		0.984
Chronic Kidney Failure-n (%)	24 (26.7%)		23 (25.8%)		47 (26.3%)		0.892
CPB time (min)	78.1 ± 32.5		77.9 ± 32.2		78.0 ± 32.3		0.967

BMI: body mass index, ICU: intensive care unit, COPD: chronic obstructive pulmonary disease, CPB: cardiopulmonary bypass. All the data are presented as medians with their respective interquartile range in parentheses. *p*-value was calculated after the comparison of the medians between the coronary artery disease group and the heart valvular disease one.

**Table 2 jpm-13-00473-t002:** Data from all the patients included in the study regarding white blood cells, leucocytes, monocytes, platelets (all expressed in ×10^3^/μL), neutrophil-to-lymphocyte ratio (NLR), and platelet-to-lymphocyte ratio (PLR) preoperatively and on postoperative days 3, 5, and 7.

N = 179	Blood Components	Median ± SD	Median	Minimum	Maximum	Normal Values
Preoperatively	Total white blood cells	7.42 ± 1.99	7.12	3.72	14.72	3.80–10.50
	Absolute neutrophil count	4.51 ± 1.71	4.10	2.20	10.50	1.60–6.50
	Absolute lymphocyte count	2.07 ± 0.73	1.90	0.46	5.94	1.50–3.60
	Absolute monocyte count	0.63 ± 0.19	0.59	0.20	1.24	0.20–1.00
	Total platelets	223.89 ± 71.53	210.00	85.00	460.00	150.00–450.00
	NLR0	2.49 ± 1.75	2.11	0.59	15.65	
	PLR0	119.26 ± 53.75	109.01	21.21	359.34	
Postoperative Day 3	Total white blood cells	13.47 ± 4.78	12.66	5.12	35.09	3.80–10.50
	Absolute neutrophil count	10.97 ± 4.18	10.40	3.90	28.70	1.60–6.50
	Absolute lymphocyte count	1.21 ± 0.56	1.10	0.04	4.06	1.50–3.60
	Absolute monocyte count	1.24 ± 0.53	1.12	0.20	3.03	0.20–1.00
	Total platelets	140.51 ± 53.94	132.00	55.00	417.00	150.00–450.00
	NLR3	10.98 ± 11.66	8.97	2.01	142.86	
	PLR3	145.09 ± 175.92	117.05	18.47	2142.38	
Postoperative Day 5	Total white blood cells	11.63 ± 3.91	11.49	3.86	25.25	3.80–10.50
	Absolute neutrophil count	19.44 ± 3.70	8.90	2.70	23.90	1.60–6.50
	Absolute lymphocyte count	1.21 ± 0.53	1.09	0.26	3.56	1.50–3.60
	Absolute monocyte count	0.80 ± 0.32	0.77	0.17	2.02	0.20–1.00
	Total platelets	135.46 ± 56.47	122.50	28.00	451.00	150.00–450.00
	NLR5	9.41 ± 6.63	7.41	2.16	55.58	
	PLR5	132.04 ± 81.73	115.59	17.61	715.87	
Postoperative Day 7	Total white blood cells	9.40 ± 3.71	8.86	3.76	33.31	3.80–10.50
	Absolute neutrophil count	6.88 ± 3.54	5.90	2.30	30.50	1.60–6.50
	Absolute lymphocyte count	1.38 ± 0.56	1.29	0.36	3.94	1.50–3.60
	Absolute monocyte count	0.86 ± 0.34	0.82	0.27	2.23	0.20–1.00
	Total platelets	190.30 ± 72.42	182.00	38.00	460.00	150.00–450.00
	NLR7	6.14 ± 5.85	4.58	1.29	58.65	
	PLR7	155.02 ± 78.75	138.19	26.21	601.43	

**Table 3 jpm-13-00473-t003:** Discrimination of 179 patients based on the number of the positive predictive factors during their first seven postoperative days.

Score	Patients n (%)	Number of Patients Who Died (% of the Respective Group)
0	28	(15.6)
1	79	(44.1)
2	46	(25.7)
3	26	(14.5)

**Table 4 jpm-13-00473-t004:** The results of the uni- and multivariate analysis for identifying potential independent risk factors for mortality.

Type of Analysis	Univariate			Multivariate		
Independent Variable	OR	95% CI	*p*-Value	OR	95% CI	*p*-Value
Gender (F/M)	0.323	0.040–2.604	0.289	2.264	0.090–57.017	0.620
Age (years)	1.080	0.994–1.173	0.068	0.791	0.512–1.224	0.124
MUST Score	1.334	0.517–3.443	0.551	N/I		
EuroSCORE II	1.359	1.038–1.779	0.025	1.065	0.538–2.108	0.858
Coronary Heart Disease	0.525	0.148–1.861	0.318	N/I		
Heart Valvular Disease	1.905	0.537–6.756	0.318	N/I		
DM	1.440	0.423–4.906	0.560	N/I		
COPD	0.888	0.226–3.488	0.865	N/I		
CKD	1.905	0.537–6.756	0.318	N/I		
ICU Stay (days)	1.373	1.186–1.590	<0.001	1.361	1.045–1.774	0.022
PLR3	0.979	0.961–0.997	0.026	0.896	0.798–1.006	0.063
NLR5	1.110	1.034–1.192	0.004	0.866	0.662–1.133	0.293
NLR7	1.297	1.132–1.487	<0.001	2.143	1.076–4.267	0.030

F: female; M: male; PLR3: platelet-to-lymphocyte ratio on postoperative day 3; NLR5: neutrophil-to-lymphocyte ratio on postoperative day 5; NLR7: neutrophil-to-lymphocyte ratio on postoperative day 7; EF: ejection fraction; DM: diabetes mellitus; COPD: chronic obstructive pulmonary disease; CKD: chronic kidney disease; ICU: intensive care unit; N/I: Not included in the multivariate analysis. 95% CI: 95% Confidence Intervals, OR; Odds Ratio.

**Table 5 jpm-13-00473-t005:** Pearson correlation coefficients between PLR and NLR on postoperative days 3, 5, and 7.

N = 179	Pearson Correlation Coefficient	*p*-Value
PLR3-NLR3	0.916	<0.001
PLR5-NLR5	0.530	<0.001
PLR7-NLR7	0.418	<0.001

PLR: Platelet-to-lymphocyte Ratio, NLR: Neutrophil-to-lymphocyte Ratio. The number following NLR and PLR states the postoperative day.

## Data Availability

The data that support the findings of this study are available on request from the corresponding author GT.

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
