# Peer review of "Neutrophil-to-Lymphocyte Ratio and Platelet-to-Lymphocyte Ratio as Predictive Factors for Mortality and Length of Hospital Stay after Cardiac Surgery"

_jpm, 2023, doi:10.3390/jpm13030473_

Round 1

Reviewer 1 Report

Present study aims to assess whether NLR and PLR could predict mortality and length of hospital stay (LOS) after cardiac surgery.

These biomarkers are easily available and costless, thus their use may be helpful in the clinical practice as adjunctive tool. The topic may be of interest for JPM readers, however some points need to be further improved:

-Please, better report inclusion criteria and add exclusion criteria in the Material and Method section

-Correct typing errors along the text. Ex: Tab 1 MELETUS. / line 356 6.Patent must be eliminated by the text

- NLR and PLR are very simple data to collect. However, what they may add to other equally simple and available biomarkers (such as CRP)? How these results may facilitate the achievement of personalized medicine? Discuss these points in the discussion section

-Complete the author contribution section at the end of the manuscript

Author Response

 "-Please, better report inclusion criteria and add exclusion criteria in the Material and Method section."

In lines 59-65 in the Material and Method section, we added, as the reviewer kindly suggested, the inclusion and exclusion criteria for the sample recruitment.

"-Correct typing errors along the text. Ex: Tab 1 MELETUS. / line 356 6.Patent must be eliminated by the text."

We have corrected all the typing errors in the text, as the reviewer correctly pointed out.

"- NLR and PLR are very simple data to collect. However, what they may add to other equally simple and available biomarkers (such as CRP)? How these results may facilitate the achievement of personalized medicine? Discuss these points in the discussion section."

We thank the reviewer for this comment. In lines 374-382, we have discussed the perspectives of NLR and PLR under the prism of personalized medicine.

"-Complete the author contribution section at the end of the manuscript."

We apologize for this omission. We have completed the author contribution section at the end of the manuscript.

Reviewer 2 Report

The work in the title contains an assessment of various configurations of blood counts elements in the prediction of early postoperative death. Meanwhile, the only one preoperative measurement (NLR 0 and PLR 0) does not differ in the group of deaths and survival. Surprisingly the authors concludes that “NLR and PLR are efficient predictive factors for 90day mortality” which seems to be irrelevant.

In addition, the reported mortality rate of >6% is unacceptably high in patients with Euroscore between 2 and 3 % and EF>50%. There is no information whether the group of deceased clearly differed in any pre- or intra-operative variables. It should be stated that such a high mortality rate in the group of elective patients without giving the reasons for this mortality does not authorize research on a relatively insignificant laboratory parameter and to draw far-reaching conclusions regarding postoperative mortality.

Author Response

"The work in the title contains an assessment of various configurations of blood counts elements in the prediction of early postoperative death. Meanwhile, the only one preoperative measurement (NLR 0 and PLR 0) does not differ in the group of deaths and survival. Surprisingly the authors concludes that “NLR and PLR are efficient predictive factors for 90day mortality” which seems to be irrelevant."

We thank the reviewer for this insightful comment. Our study aimed to assess the potential predictive value of postoperative NLR and PLR values in patients undergoing cardiac surgery regarding 90day postoperative mortality and length of hospital stay. Thus, we aimed to assess whether NLR and PLR in the first 7 postoperative days could predict patients who would be at higher risk for complications and thus longer hospital stay or would die in the next three months. Preoperative values of these indices did not differ significantly between the two groups (dead vs survivals), indicating that the two groups, that would be compared, were homogenous and there was not any selection bias (e.g. a more critically ill patient with more serious comorbidities, would probably have a worsened immunological status reflected by NLR and PLR, and thus it would be more possible to die after cardiac surgery). However, postoperative (in the first 7 days after surgery, measured every 2 days after the 3rd postoperative day) NLR and PLR values were found to be predictive for 90day mortality and length of hospital stay. This was the reason why we concluded that “NLR and PLR are efficient predictive factors for 90day mortality”.

"In addition, the reported mortality rate of >6% is unacceptably high in patients with Euroscore between 2 and 3 % and EF>50%. There is no information whether the group of deceased clearly differed in any pre- or intra-operative variables. It should be stated that such a high mortality rate in the group of elective patients without giving the reasons for this mortality does not authorize research on a relatively insignificant laboratory parameter and to draw far-reaching conclusions regarding postoperative mortality."

We thank the reviewer for this comment. Regarding the differences between the groups of survivals and non-survivals, we have already reported that they differ regarding the duration of remaining intubated postoperatively (lines 137-142). There was not any difference in any preoperative or intraoperative variable, based on the fact that the same surgical technique was performed by the same surgical team. Regarding mortality, 30day mortality (6 out of 179 patients) was 3.35% and 90day mortality was 6.15%. These percentages are equivalent to the percentages presented in the original EuroScore II study (4.048% and 6.023%, respectively) [Nashef, Samer A M et al. “EuroSCORE II.” European journal of cardio-thoracic surgery: official journal of the European Association for Cardio-thoracic Surgery vol. 41,4 (2012): 734-44; discussion 744-5. doi:10.1093/ejcts/ezs043]. We discussed this issue in lines 387-393.

Round 2

Reviewer 1 Report

Authors answered satisfactorily to points raised by the reviewer 

Author Response

We thank the reviewer for her/his comments.

Reviewer 2 Report

The authors' explanations are seemingly convincing, but I did not notice any significant changes in the text. Thus, the work is still inconsistent. Basic operational data is still missing. It is known that the course of surgery can affect postoperative laboratory parameters, and not vice versa. There is still no comparison of basic data like ECC Time and X clamp time. Some statements in the paper suggest that the authors underestimate the factor of surgery in the prognosis; in the section „Introduction” in line 42, the authors write: „Patients undergoing anesthesia for cardiac surgery experience a systemic inflammatory response”. I believe, that anestesia itself does not influence very much on inflammation.

Author Response

We thank the reviewer for these comments. We agree that the course of the surgery can affect postoperative laboratory parameters. Data regarding the X clamp time were not recorded. Regarding ECC, we added this information in lines 136-138. It is worth mentioning again that the surgical and anesthesia team was exactly the same in every case and performed the same procedure following the same surgical technique. Additionally, we did not underestimate the effect of cardiac surgery on the prognosis, but both survivals and non-survivals experienced the same surgical stress and the mortality rate of such procedures is already known. Regarding what is written in line 42, we reported that “Patients undergoing anesthesia for cardiac surgery experience a systemic inflammatory response more aggravated” compared to other surgeries (due to the additional stress of extracorporeal circulation). This does not mean that the procedure does not add surgical stress, but that in cardiac surgery the stress of ECC during the procedure contributes to additional inflammatory response than other types of surgeries. Finally, we have also mentioned the effect of anesthesia on NLR in lines 302-304.